

# The Delphi method to analyze the expert views on possible futures of the smart city adoption and development in developing countries: the case of Jordan

Muneer Nusir[1], Mohammad Alshirah[2], Sahar ALMashaqbeh[3], Mohammed Yousuf uddin[1], Sultan Ahmad[4] and Sana Fakhfakh[1,5]

[1] Department of Information Systems, College of Computer Engineering and Sciences, Prince Sattam bin Abdulaziz University, Alkharj, Riyadh, Saudi Arabia
[2] Information Systems Department, Al al-Bayt University, Mafraq, Jordan
[3] Department of Industrial Engineering, Faculty of Engineering, The Hashemite University, Zarqa, Jordan
[4] Department of Computer Science, College of Computer Engineering and Sciences, Prince Sattam bin Abdulaziz University, Alkharj, Riyadh, Saudi Arabia
[5] MIRACL Laboratory, Sfax University, Sfax, Tunisia

Corresponding author
Muneer Nusir,
moneer.techno@gmail.com

## ABSTRACT

Smart cities are characterized by the integration of various technologies and the use of data to achieve several objectives. These objectives include the creation of efficiencies, boosting economic development, expanding sustainability, and improving the overall quality of life for individuals residing and working within the urban environment. The aim of this study is to analyze the future of smart cities with respect to developing countries, specifically Jordan as the case. This analysis is based on the opinions and feedback from the field experts. In this study, we are tapping into multiple domains of smart cities such as smart governance, education, healthcare, communication, transportation, security, energy, and sustainability. The field experts' consensus was developed with the Delphi method. The Delphi survey comprises eight questions to assess the views about smart city adoption and development with respect to Jordan. The results and findings of this study revealed specific challenges and opportunities in smart city adoption with respect to Jordan. The experts' opinions have validated the study of the 2023 Smart City Index report. They have offered crucial input and future guidance for the adoption of smart cities in Jordan. Additionally, they have indicated which domains of smart cities should be prioritized during the implementation in Jordan.

## INTRODUCTION

The global smart city movement has grown dramatically during the past ten years. Global momentum for the construction of smart cities has been building since the early 2000s (*Li, Taeihagh & Tan, 2022*; *Lim et al., 2021*). To date, there is not a standard or agreed definition of a "smart city" (*Ruhlandt, 2018*; *Dashkevych & Portnov, 2022*; *Myeong, Park &*

*Lee, 2022*). Despite the disparity in the definition of a smart city, the European Commission (*European Commission, 2022*) defines a smart city as "a place that integrates physical," digital and human systems in traditional networks and services to better use energy resources and reduce emissions to the benefit of citizens and businesses". There are many different domains that smart cities may fall under and interlink to, yet both academics and practitioners share certain perspectives. According to *Giffinger et al. (2007)*, the main six domains that are employed in a city's smart city performance: smart mobility, people, living, governance, environment, and economy. The two common methodologies used in smart city research are either technology-oriented or people-oriented (*Bibri & Krogstie, 2017*) most recent research (*Yigitcanlar et al., 2022*; *Li et al., 2022*) on smart cities has acknowledged that the strategy should be people-oriented rather than technology-oriented. Smart cities are viewed as multifaceted systems that combine economic, social, and physical capital with Information and Communication Technology (ICT) infrastructure to improve a city's intelligence (*Silva, Khan & Han, 2018*). To improve the sustainability and livability of cities, it is important to seek smart city development to discover creative answers to diverse social, economic, and environmental concerns. As a result, the construction of a smart city necessitates the participation of several stakeholders and economic sectors at various governmental levels (*Silva, Khan & Han, 2018*; *Anthopoulos & Kazantzi, 2021*).

*Berrone & Ricart (2020)* rated 101 indicators utilized in smart cities throughout 10 major aspects, the Cities in Motion Index had established, compromising technology, urban planning, the environment, mobility and transportation, governance and public management, international projection, social cohesion, human capital, and the economy. After assessing 174 cities based on the earlier criteria, they found London is ranked number 1 as the smartest city worldwide, further stated, the discrepancy between developed and developing nations in smart city development, none of the top twenty smart cities are from developing countries (*Berrone & Ricart, 2020*). However, there are very few research studies were conducted on evaluating smart city adoption, except for researchers such as (*Lim et al., 2021*; *Bhattacharya et al., 2018*; *Darmawan et al., 2020*) who evaluated smart city adoption and acceptance in some cases from developing countries broadly and the obstacles it faces. Therefore, the authors endeavor to uncover the current knowledge gaps in practice and address the following questions:

- What level of adoption and/or adaptation do experts have of the smart city for applicable transformation in Amman (Jordan's capital city)?
- What level of understanding do experts have of the smart city in regard to the challenges facing smart city development in Amman (Jordan's capital city)?

Based on these research questions, this study aims to examine the factors that are pressing infrastructural needs based on experts' experience and knowledge to adopt smart city services in developing countries; to meet the growing demands in urban areas. The main purpose of this study is to ascertain the levels of public and/or specific knowledge and adaptability in order to provide governments and policymakers with direction on how to enhance smart city plans and initiatives and make more intelligent and inclusive living options available to their citizens. The findings of this study are reproducible and applied

to other developing countries, which have the same context to provide a comprehensive picture of the crucial factors affecting citizen acceptance of smart city systems in these regions.

The subsequent sections of the article are organized in the following manner: the 'Literature Review' section outlines the literature review, whereas the 'Research Methodology' section explains the proposed methodology. The 'Results and Analysis' section includes results and analysis to validate, and the 'Discussion' section has a discussion of the progress of the suggested solution. 'Conclusions' offers a conclusion as a comprehensive summary of the study and outlines its potential for further development.

## LITERATURE REVIEW

The works that might be regarded as relevant to our field of study are discussed in this section. Significant progress has been made in the field of smart cities in recent years with the intention of raising people's quality of life. To do this, technologies (*Uddin & Ahmad, 2020*) like artificial intelligence (AI), edge computing, and the Internet of Things (IoT) are being used.

In a work related to smart cities, *Munawar et al. (2022)* conducted a review in a study on smart cities that looked at how these cities could be built to take use of disruptive technology and enhance post-disaster management. On the other hand, a review of video structural analysis in connection to smart transport was published by *Zhao et al. (2021)*. *Toh & Milojicic (2021)* provided a review of the Visual IoT work that has been done in the area of smart cities, but they avoided going into specifics about the methodologies or performance metrics. *Desislavov, Martínez-Plumed & Hernández-Orallo (2023)* in their study, looked at the connection between deep learning's influence on energy consumption in smart cities and the exponential expansion of AI parameters in general.

To examine the effect of security-related factors in predicting intention to use smart city technology, *Grandhi, Grandhi & Wibowo (2021)* provided the Sec-UTAUT model. The knowledge domains of technology adoption and smart cities are enhanced by this study. The findings of this study can be used by city governments to create suitable security measures and hasten the adoption of smart cities. This study's flaw is the absence of empirical support.

In a different work (*Abu Salim et al., 2021*) writers looked at how smart city services (SCS) are delivered as well as the demographic and behavioral traits of users that affect their intention to adopt as well as their contentment or dissatisfaction with the services. One of the initial research looking into the causes of SCS usage patterns in the Middle Eastern region is this one. Their findings have repercussions for both the advancement of theoretical knowledge and actual managerial implementation.

In a study by *Habib, Alsmadi & Prybutok (2020)*, they apply empirical analysis to identify the characteristics that have the greatest impact on citizens' and government employees' intentions to use smart-city services. A mid-sized U.S. city is used as a case study for the development and testing of a Smart Cities Stakeholders Adoption Model (SSA), which is based on the Unified Theory of Acceptance and Use of Technology (UTAUT2). They also

show that perceived privacy and security are significant predictors of trust in technology, and pricing value is a predictor of trust in government. These results provide municipal authorities with a method for assessing residential desire to use smart city services and identify the elements necessary to create a winning smart city strategy.

A framework and study that analyses the connection between gender, security, and the smart city have been offered by *Bansal et al. (2021)*. The initial goal is to conceptualize security and describe how a smart city might offer a solution to the noted issue. They conclude that because safety and security have different meanings, they shouldn't be used interchangeably. Different people and places will experience safety and security in different ways.

The term "smart city" has become more popular recently, while urban science and the use of rational city management techniques have been studied for a long time (*Schultz & McShane, 1978*). Despite this, there is still no broadly accepted definition of this new idea of the "smart city" among academicians and practitioners (*Chourabi et al., 2012*). The hazy concept of "smart cities" aims to use massive data streams (big data) collected from society as a way to justify the necessity for intelligent services in our communities.

Smart cities around the world are few with unique focus and initiatives. Singapore is considered as the leading smart city in the Asia with wide range of initiatives, including smart traffic management, smart energy grids, and e-government services (https://www.smartnation.gov.sg/). Abu Dhabi and Dubai are the top smart cities in the Middle East region, leveraging technology to improve government services, infrastructure, Autonomous Transportation Strategy, which aims to develop and implement self-driving vehicles and other autonomous transportation solutions to improve quality of life (*United Arab Emirates, 2024*).

The growth and development progress of Jordan's smart cities and how it stacks up to other developed countries is studied. According to *O'Neill (2023)*, 91.83% of Jordan's population was urban in 2022. From the middle of the nineteenth century, this percentage in Jordan seems to grow ferociously. As a result, Jordan's unexpectedly rapid urbanization comes with many issues. Some of these issues are the same as those in developing countries (*Nusir, Alshirah & Alghsoon, 2023*), like poverty, diversity, pollution, and power outages, deteriorating infrastructure, defamation of the government and bureaucracy, lack of public participation, and poor moral quality. Thus public institutions, NGOs and other Jordanian institutions struggle to provide programs to address the major issues of urbanism as part of smart city programs. In a study of UN in Jordan, the spatial distribution of population, Jordan refugee population, displacement dynamics in Jordan, governance and administration in Jordan, land cover and flood vulnerability are the major constraints in smart city transformation and towards the achievement of the sustainable development goals (*UN-Habitat, 2023*). There are other obstacles facing the idea of a smart city in Jordan as well. The degree to which local residents, businesses, and visitors engage in energy-saving and technologically-driven activities, for instance, varies. To ensure that this vision becomes a reality, new information and communication technology standards, infrastructure, and solutions are needed for the smart city concept to be implemented. The

progressive integration of ICTs as a means of facilitating innovative approaches to urban problems in urban areas will help towards smart city transformation.

Cities in developing nations struggle with serious concerns like poverty, pollution, unemployment, diversity problems within one city, a low standard of living, congestion, and immigration (*Toh & Milojicic, 2021*). On the other hand, they can help them develop their cities by implementing smart city solutions based on modern technologies. There have been several models put up for the implementation of smart cities that seem appropriate for the situations in developed nations. As a result, it is essential to construct a framework that takes into account the unique circumstances of developing nations. They can address their particular problems with the help of this framework (*Abu Salim et al., 2021*).

The following are the main factors affecting smart city development, especially in developing countries like Jordan.

- The high setup costs of a smart city include the infrastructure and superstructure requirements.
- The low degree of computer literacy among the city's residents is a concern for the smart city. The local community needs to be ready for the concept of a smart city to take off.
- Issues with security, cookies, computer infections, and privacy violations (spam and junk mail).
- The challenge of creating a thriving information society in the urban area. Numerous digital natives contend with aliases and concealed personas, warping authentic communication and diminishing the social influence within the shared data.
- Lack of comprehensive plan and a well-defined digital transformation strategy. City planners and administration showed little interest in or knowledge of the smart city concept, which is a subset of e-government.
- The disparities in spatial databases pertaining to the elements of attractions and the fragmentation of efforts among multiple stakeholders to reach the smart city.
- Inability to translate strategy and vision into initiatives and goals that can be carried out.

In the context of smart cities, ICT Infrastructure is a very wide topic and includes most aspects of ICT, both hardware and software like network infrastructure, software applications, cloud computing, data platforms and access devices. Also numerous studies demonstrate that residents with higher ICT proficiency and experience will benefit more from the smart city transformation than residents with lower proficiency and experience (*Habib, Alsmadi & Prybutok, 2020*; *Grandhi, Grandhi & Wibowo, 2021*).

ICT and the digital economy are two trends that have become more prominent in the twenty-first century, impacting each nation's yearly GDP growth. The complexity, diversity, end-user devices, and Internet of Things (IoT) connectivity of ICT infrastructures increase day by day. Furthermore, the applications push large volumes of data, demand higher bandwidth, and are more interactive in order to support real-time analysis and solutions in the rising field of ICT adoption towards smart city development. For the purpose of digital transformation, current information and communication technologies (ICT) and their convergence and effects on the modern world should be thoroughly examined. It

focuses on the analysis of AI and how it may optimize ICT functions overall and assist workers in making better decisions more quickly in the new digital world (*Desislavov, Martínez-Plumed & Hernández-Orallo, 2023*; *Munawar et al., 2022*).

Additionally, it evolves into the confluence of edge computing, big data, and AI in the IoT, which connects intelligent sensors and actuators. They enable real-time data generation, collection, and processing in data centers (*Bansal et al., 2019*). Therefore, in light of the new requirements and necessities for the development of ICT ecosystems for the development of smart cities, a thorough analysis of data and their evolution towards the edges and the deployment of SDN networks are also required.

## RESEARCH METHODOLOGY

The aim of this study is to identify and assess the possible future of smart city adoption and development in Jordan. This study followed a qualitative method because it explored the views of experts in smart cities and their sub-domains like transportation, communication, energy, health, and research. Field experts are personally contacted to contribute their views and suggestions on multiple challenges and opportunities in the area of smart city development. The methodology may possess subjectivity and intuition, making it suitable for complex investigations requiring a human perspective. The Delphi methodology is used to conduct the survey and collect the opinions and views of the selected experts. The survey is conducted through Microsoft Forms. Survey data is further analyzed to find major challenges in the implementation of smart cities. The survey consists of eight questions that require a descriptive answer. The survey focuses on potential areas for smart cities and obstacles to adopting smart cities, including areas that require more research and development. Further, our survey focuses on how smart cities are going to address the sustainability, energy, mobility, and privacy issues of citizens. The results collected are grouped according to similarities in feedback and contrasting opinions. This study first step starts with identifying the smart city experts from multiple domains like smart governance, academic research, healthcare, communication, transportation, security, energy, and sustainability, shortlist the potential participants for the study and establish communication with the shortlisted experts. Step two an online form is created with eight questions, share the link with participants to fill the form with their opinions on different questions. Step three responses were collected, duplicated responses were removed and a qualitative analysis with illustrations and tables is created. Figure 1 illustrates the steps and activities performed in this project. In this kind of research, it is up to the experts to do an in-depth analysis of the elements that form the basis of the topic being researched and to offer their opinion on the basis of the vast amount of relevant experience that they have accumulated over the course of their careers (*Sourani & Sohail, 2015*). The availability of a number of experts in smart city development is limited. Delphi expert panel size could be modest with participants from 10 to 18 (*Okoli & Pawlowski, 2004*). In this study panel size was 18 and responses from experts were overwhelmingly good.

Delphi method is proved to be reliable measurement instrument to forecast or decision-making. Delphi method arrive at a group opinion or decision by surveying a panel of

**Step 1**
- Identify smart experts
- Shortlist experts to best fit the study
- contact experts

**Step 2**
- Formulate Questions
- Develop Online form
- Share the link with experts

**Step 3**
- Collect the responses
- Remove duplicates
- Categorize responses based on smart city domains
- Qualitative Analysis of Expert Responses

**Figure 1** **Steps in collecting and analyzing opinions of smart city experts.**

experts. Questionnaire sent to a panel of experts so they can reach a consensus after multiple rounds. Results are presented with an aggregated summary. The Delphi process has no specified number of rounds in literature as previous studies have adopted the best-fitting approach. However, various studies, in the field of engineering management, have performed the two or three rounds before reaching a consensus (*Ameyaw et al., 2016*). There is no guarantee that all participants will complete the survey completely, but a reasonable number of participants will complete the survey (*Chan et al., 2001*).

The selection of Delphi panels is one of the important tasks at the beginning of this research study. The survey's questions can be answered with feedback that is correct, authentic, and dependable if appropriate field specialists with experience contribute to the discussion. The findings of this survey give a more accurate picture of the opportunities and problems involved in building smart cities in Jordan, specifically as a result of their findings. Within the scope of this study, participants included academics, researchers, and industry professionals. The experts that were chosen for this investigation have substantial prior experience working in any of these sectors of the smart city. Participants in this study

were chosen to be experts in a variety of subfields pertaining to the implementation of smart cities. Priority will be given to individuals who are currently active in the field of smart cities. These specialists are currently employed in different locations in Jordan. Before deciding whether or not to invite someone to take part in this study, their profile is given careful consideration. It was attempted to get in touch with around twenty specialists in the field by means of e-mails and telephone calls; of these, eighteen answered favorably and gave their valuable feedback by answering the given questionnaire. Experts were contacted and briefed on the study and its goals before completing the survey to ensure that they would provide accurate responses. The Delphi approach is effective in situations where the topic being addressed may gain value from the collective input of subjective judgments or conclusions and where the dynamics across a group limit effective communication (*Grime & Wright, 2016*). The Delphi method is a structured approach used to get expert opinions in a systematic way. The goal is to get a panel of experts selected for participation in the process and arrive at a reasonable consensus. There is no specified number of iterations for the Delphi process (*Sourani & Sohail, 2015*). However, based on the availability and participation agreement with experts, it will be decided how many iterations are possible. In this study, a two iterations are completed with eight critical questions listed in Table 1. In this study, a Microsoft form with eight questions is given to selected experts in smart cities, results are shared with the experts in round 1 and participants reach a consensus in round 2 without any changes in their feedback. Further results are exported as a spreadsheet for further analysis.

# RESULT AND ANALYSIS

In this study after collecting the feedback from smart city experts, out of 18 participants it was the academic research domain who participated the most *i.e.,* 39%, energy efficiency, others, smart city project management, smart transportation participation is 11% each and data analyst, sustainability coordinator and urban planning participation is 6% each, participation is show in Fig. 2. In this survey participants are free to any number of questions from the eight questions in the form and finally a consent is take from the participants to use this data for this study and any other study we undertake in future. In the remaining section of this article we analyze and discuss each question individually.

Question 1: Which smart city technologies do you believe have the greatest potential for success in the near future?

Technologies identified by our experts are the enablers for building smart cities. Problem driven approach is a successful method for starting smart city development, where the problems of the residents of the city are addressed through smart city technology (*Komninos, 2018*). Most of the experts in this study are concerned with sustainable green infrastructure, transportation is given much emphasis, which presumes that it is one of the problems with which citizens are living in Jordan, and Fig. 3 illustrates the responses for question one in word cloud. Integrating artificial intelligence (AI) with information and communication technology (ICT). Integration of data management networks, such as the Internet of Things (IoT), big data, and cloud computing technologies is required for sustainable

**Table 1   Questions related to the possible future of the smart city adoption and development.**

| S.No | Question |
| --- | --- |
| 1 | Which smart city technologies do you believe have the greatest potential for success in the near future? |
| 2 | What, in your opinion, is the greatest obstacle to the widespread application of smart city technology and other critical issues facing smart cities today? |
| 3 | How do you think smart cities can best address issues related to energy efficiency and sustainability? |
| 4 | What are the most important factors cities should consider when developing and implementing smart city technology? |
| 5 | How do you think smart cities can best address issues related to transportation and mobility? |
| 6 | How do you believe smart cities may effectively address concerns relating to citizen engagement and participation while maintaining the privacy and security of their residents? |
| 7 | How do you think smart cities can best address issues related to affordability and equity? |
| 8 | What are the most important areas of research and development that should be prioritized in smart cities? |

economic growth and the administration of available resources through participatory governance (*Voda & Radu, 2018*). There are a range of ever evolving technologies, like IoT, blockchain, 5G, 6G networks, robotics, Industry 5.0 are congregated to build and run sustainable smart cities (*Javed et al., 2022*). Communication technology is the backbone of the smart cities, to achieve high data rate communication along with low signal attenuation, efficient spectrum utilization, high scalability, coverage at low cost and agile encryption mechanisms. The emerging networking and communication technologies like software defined wireless networking (SDWN), network functions virtualization (NFV), visible light communication (VLC), cognitive radio networks (CRNs), green communication (GC), 6LowPAN, Thread (IP-based IPv6 networking protocol), Sigfox, Neul, and NFC hold the potential to become a major force in achieving seamless connections within smart city ecosystems (*Yaqoob et al., 2017*).

Question 2: What, in your opinion, is the greatest obstacle to the widespread application of smart city technology and other critical issues facing smart cities today?

In this study, most of the experts were concerned with multiple issues as shown in Fig. 3. If broadly categorized, the issue may fall into one of the following categories: governance, citizen engagement, infrastructure, or data. Smart city development is a state-level decision and responsibility. The government develops strategies. Goals and objectives are defined, along with the initiative to achieve these goals. In the development phase, the government is responsible for initiating policies and regulations to support innovation and the adoption of emerging technologies, which are the enablers of smart city development. A wide range of stakeholders are part of smart city development, such as government agencies, business enterprises, research labs, academia, and the public. The participation of the public in the

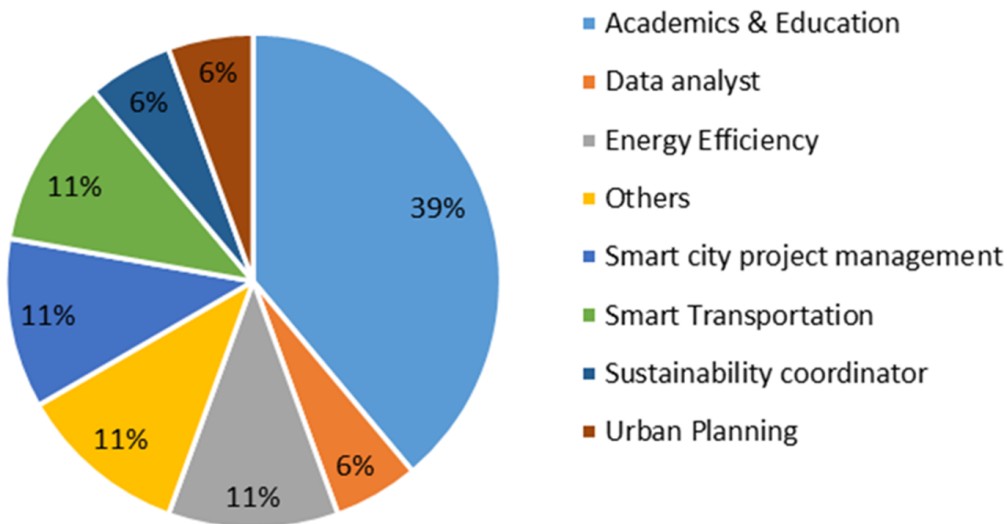

**Figure 2  Expert participation.**

decision making process is crucial to the success of smart city implementation. Education and training of citizens to leverage smart city technologies and raise awareness about data privacy and security (*Meijer & Bolívar, 2016*). It is crucial to engage with citizens in the early stages and throughout the implementation process to build trust and support because smart city projects can raise concerns about privacy, security, and equity. This can be achieved through transparency, sharing information, and collecting feedback from the public. Accountability is also one of the cornerstones with respect to citizens where public fund data is used. This article is mainly focused on collecting potential information about smart city development from experts in the field. Information and communication technologies empowered by IoT enhance the performance of regular city operations with minimal human intervention and improve the quality of service to citizens (*Harrison et al., 2010*). The requirements for smart city development are very complex. To successfully implement smart cities, it is inevitable to collaborate with vendors, technology experts, and other stakeholders to develop and deploy smart city solutions that are affordable, reliable, and easy to use. Technologies like 5G and 6G networks, IoT infrastructure, and blockchain are difficult to some extent. Governance should address the acquisition and deployment of these technologies (*Balfaqih & Alharbi, 2022*). Data is the backbone for running smart city services effectively and efficiently. Data collected from sensors and cameras helps the transportation services manage traffic jams, provide emergency services, and optimize the movement of citizens to reduce costs and promote sustainable living. Energy consumption can be reduced with smart energy equipment and meters to reduce carbon footprints. Waste management will be effective with the use of dumpster sensors, optimizing disposal routes, reducing waste production, and increasing recycling rates (*Gharaibeh et al., 2017*).

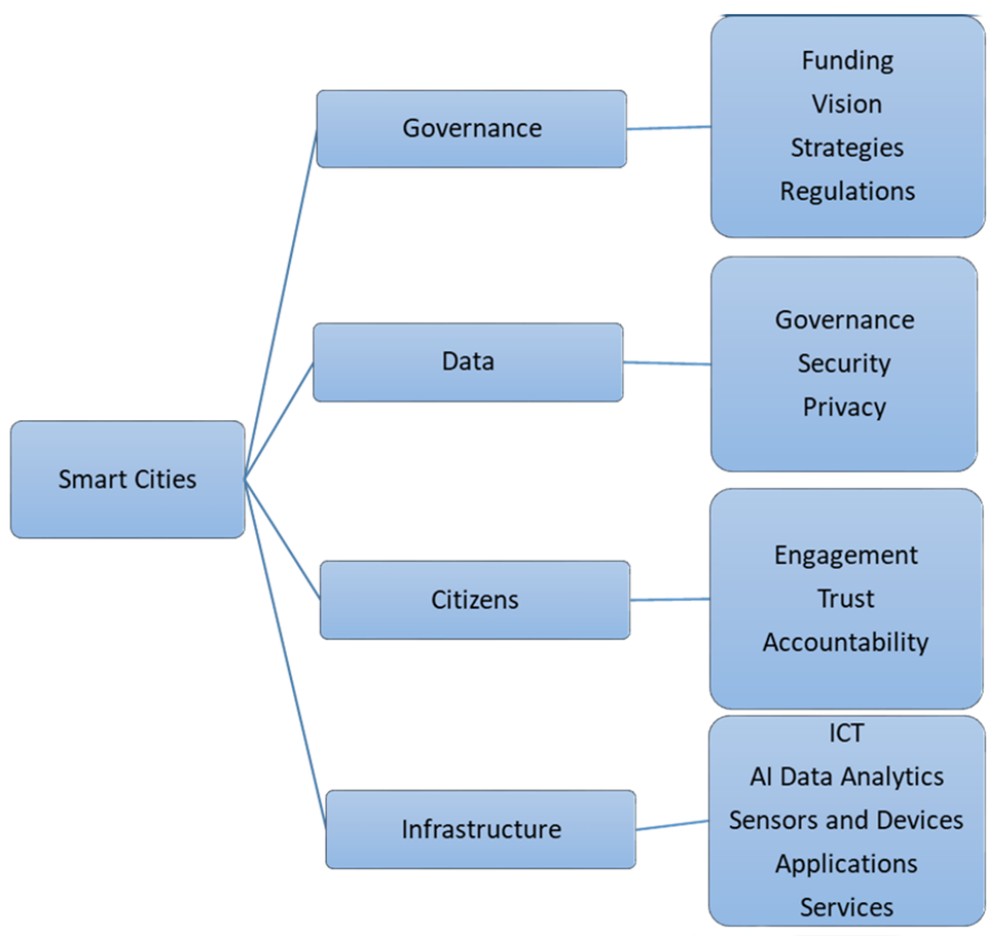

**Figure 3  Categories of issues in smart cities.**

Robust data management strategies should be in place to collect, store, process, and share information at multiple levels and service sectors, along with data integrity and security.

Question 3: How do you think smart cities can best address issues related to energy efficiency and sustainability?

Smart cities have set one of their primary objectives as achieving sustainability, and smart technologies offer a number of ways in which this can be done, including the following: sustainable transportation, reducing energy consumption, waste reduction, and conservation of resources. Smart technologies can be effective in monitoring and reducing pollution. For example, smart air quality sensors can provide data related to air pollution and alert people to areas with high pollution levels, and effective traffic management can reduce traffic jams and provide emergency services. Smart water meters can identify leaks and other inefficiencies in water supply (*Caragliu, Del Bo & Nijkamp, 2011*).

Energy efficiency and reliability in smart cities are achieved through the implementation of smart grid digital technology in the electrical grid. Smart meters, for instance, can show people how much energy they are using in real time, which can help them find ways to save

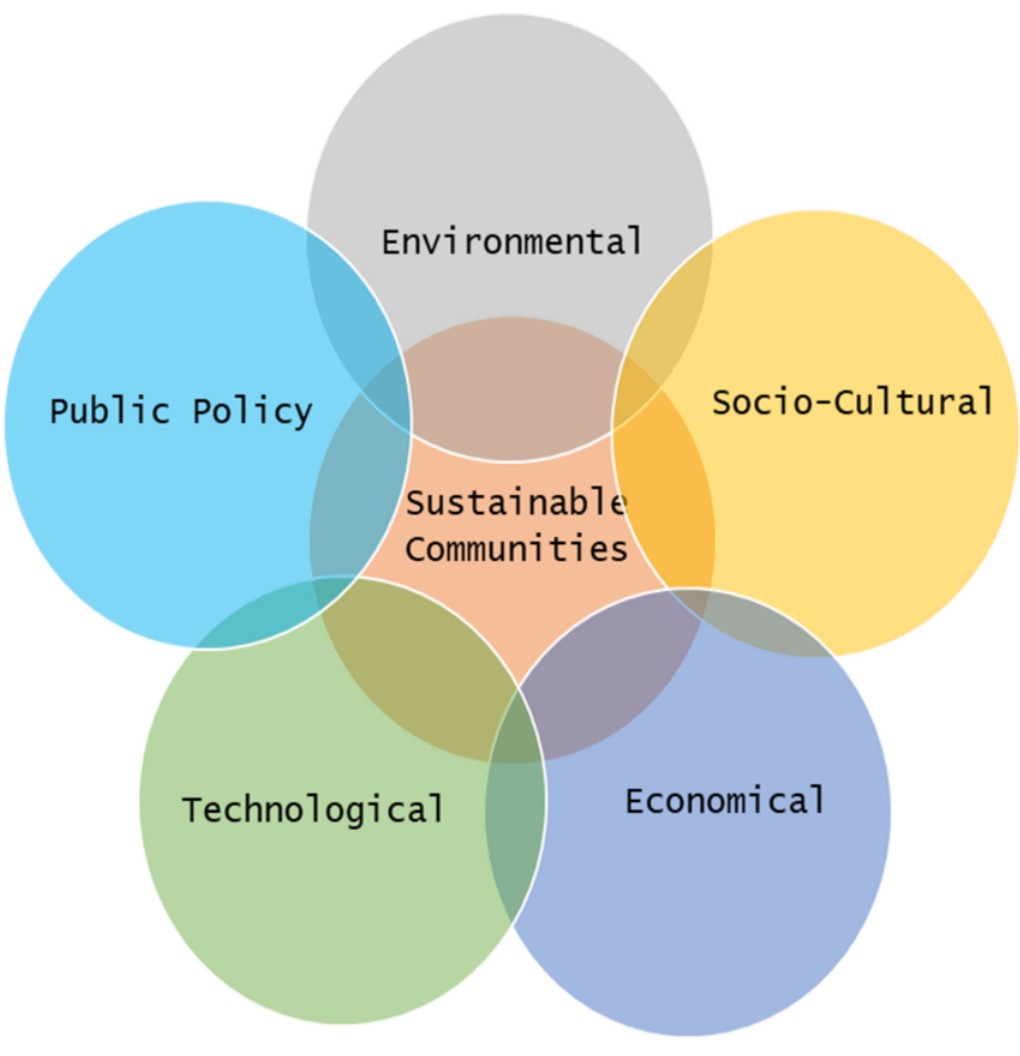

**Figure 4  Sustainability domains.**

energy. Smart grid technologies can also aid in the management of peak energy demand and the incorporation of renewable energy sources into existing electrical grids (*Silva, Khan & Han, 2018*).

The economy as a whole may benefit from sustainability efforts as they foster expansion and stability. Sustainability creates new jobs in the fields of renewable energy, waste management, and more. Public health will have a positive impact by reducing pollution and creating a clean environment, which is achievable through sustainability initiatives. The sustainability domains, as shown in Fig. 4 are essential to building sustainable communities. These domains represent the 17 sustainability development goals of the United Nations (*UNECE, 2016*).

Question 4: What are the most important factors that cities should consider when developing and implementing smart city technology?

Factors that should be considered while developing and implementing smart city technologies are identified and grouped into internal factors and external factors (*Myeong, Jung & Lee, 2018*). Citizen participation, leadership, and infrastructure are the internal factors, and the industrial revolution, political will, and stakeholders are the external factors. Citizen participation is discussed in question 6. The leadership of the local government heads highly influences the success of policy implementation (*Washburn et al., 2009*). Chief Information Officers (CIOs) in the era of smart cities must demonstrate leadership with ICT expertise for the sake of successful smart city development, including skills in long-term financing, an appropriate allocation of expertise, employee education, staff accountability standardization, and interoperability of systems (*Erboz, 2017*). There are many elements of smart city ecosystem which should be taken in to consideration which are illustrated in Fig. 5. In this study out experts have identified most of these factors to consider while developing smart cities.

Information and communication technologies (ICT) play a crucial role in connecting smart city resources, securing the management of the enormous amounts of data generated, and providing the necessary services. Smart cities are built with current state-of-the-art technology, especially information and communication technology (ICT). The objective of this progressive technology is to enhance efficacy and connectivity, targeting the long-term resilience, productivity, and wellbeing of the community. The UNECE and ITU have defined the smart sustainable city by combining sustainability principles with urbanization's increasing reliance on technology to achieve the following: "A smart sustainable city is an innovative city that uses information and communication technologies (ICTs) and other means to improve quality of life, efficiency of urban operation and services, and competitiveness, while ensuring that it meets the needs of present and future generations with respect to economic, social, environmental, as well as cultural aspects" (*Myeong, Jung & Lee, 2018*).

Smart cities need smart industries. Industry 4.0 is the development of highly automated industries through interaction between humans and machines. In this context, Industry 4.0 will inform future business models driven by state-of-the-art technologies. Industry 4.0 transforms the manufacturing and production processes in industries, resulting in a staggering effect. With the help of the IoT and cyberphysical systems (CPS), Industry 4.0 will play a significant role in transforming traditional factories into smart factories (*Oubbati et al., 2017*).

Global governments are eager to join the smart city revolution in order to provide citizens with a sustainable and cost-effective lifestyle. As new initiatives and programs are launched across the majority of the GCC, the Middle East is ramping up its smart city ambitions and investments. Since its inception, Saudi Arabia's NEOM project has been extremely ambitious in both design and scale, with a projected development cost of $500 billion and further consideration of implementing 100% renewable energy. It will be the world's top city to achieve such a feat. Dubai has already delivered the first fully 3D-printed office and the first 3D-printed laboratory; it is now time to test the technology's capacity to create a livable and welcoming home (*Menouar et al., 2017*) Stakeholders and their roles are discussed in question 2.

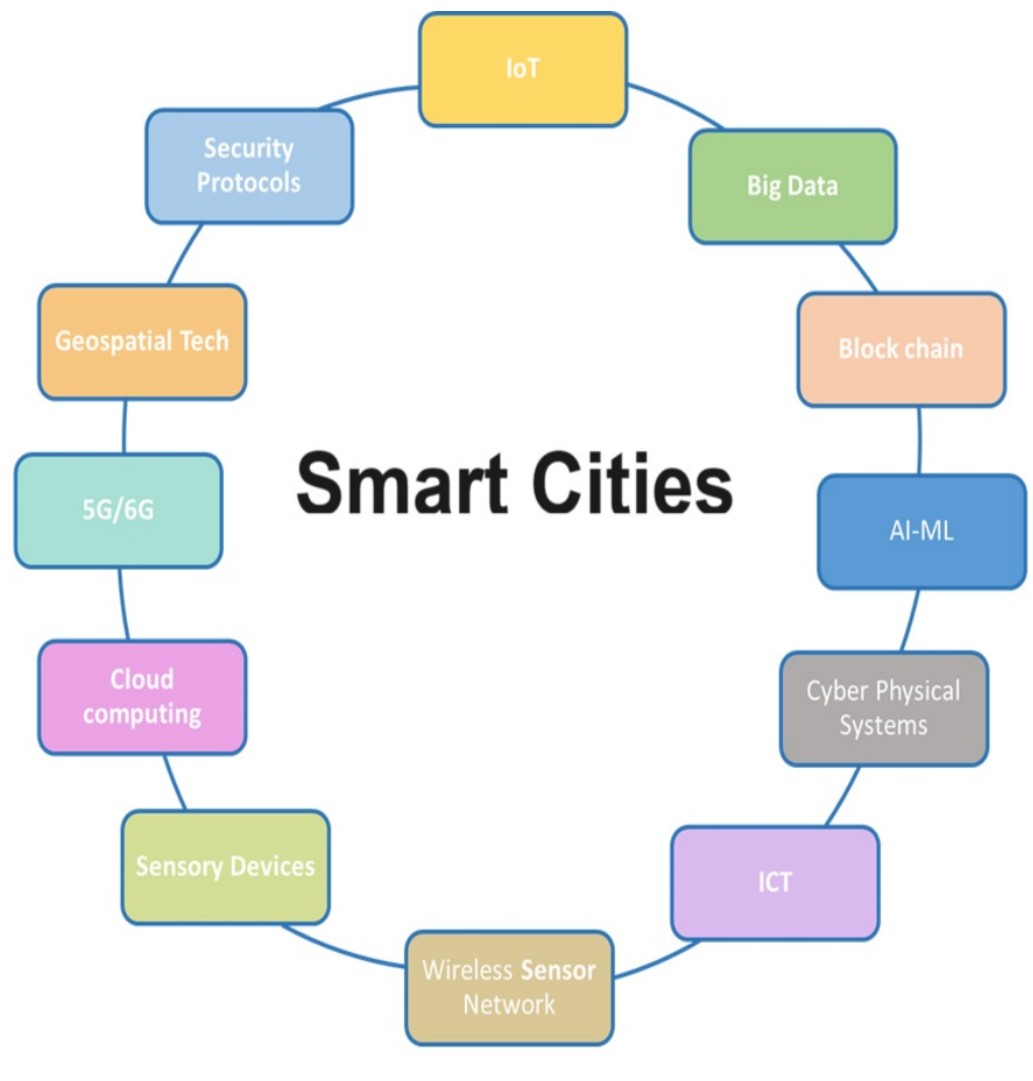

**Figure 5** Elements of smart cities.

Question 5: How do you think smart cities can best address issues related to transportation and mobility?

Smart transportation or intelligent transportation system is the inevitable part of the smart city. Smart transportation integrates multiple technologies like IoT sensors, 5G networks, wireless sensor networks, with connected and autonomous vehicles. According to US department of transportation "Intelligent Transportation Systems (ITS) apply a variety of technologies to monitor, evaluate, and manage transportation systems to enhance efficiency and safety". Unmanned Ariel Vehicles (UAV) also known as drones; UAVs deployed in military for many years. UAV's commercial use is underway. UAV has been in use of other fields such as precision agriculture, security and surveillance, and delivery of goods and services (*IMD, 2023*). The proliferation of connected and autonomous vehicles will enable the next generation of ITS technology. Connected vehicle technology provides

**Table 2   SCI scores for Jordan.**

| Structure and Technologies | Score on scale of 100 points |
|---|---|
| Traffic congestion is not a problem | 8.1 |
| Public transport is satisfactory | 30.4 |
| Finding housing with rent equal to 30% or less of a monthly salary is not a problem | 22.5 |
| Air pollution is not a problem | 29.0 |
| Green spaces are satisfactory | 27.6 |
| Residents contribute to decision making of local government | 15.3 |
| A website or app allows residents to effectively monitor air pollution | 26.4 |
| Car-sharing apps have reduced congestion | 36.7 |
| The city provides information on traffic congestion through mobile phones | 47.1 |
| IT skills are taught well in schools | 41.4 |

**Notes.**
SCI, Sensitive Compartmented Information.

vehicles on the same road with the means to communicate and exchange real-time data that can be used to improve safety (*Johnson, Acedo & Robinson, 2020*). In question 1 most of the experts have suggested transportation as the potential requirement with respect to Jordan. IMD smart city indexing (SCI) for the year 2023, which is based on human development index data provided by the United Nations Development Program (UNDP), ranked Jordan HDI 0.72 and Amman HDI 0.737. Amman city of Jordan ranks 135 in the smart city index. In SCI on the scale of 100, Traffic congestion is not a problem scored 8.1 and the public transport is satisfactory scored 30.4, which confirms our experts participated in this study also pointed out the key issues. In Table 2 represents the SCI score for the areas of smart cities discussed in this article. Priority area in SCI survey shows 64% voted for road congestion and 33% voted for public transportation (*Cortés-Cediel, Cantador & Bolívar, 2021*). In Table 3 responses from the participants in this study are shown, in which transportation and energy sectors were more emphasized.

Question 6: How do you believe smart cities may effectively address concerns relating to citizen engagement and participation while maintaining the privacy and security of their residents?

Smart cities require the participation of citizens to best capture their needs and to plan and develop smart cities that provide sustainable living, a better quality of life, and well-being. Apart from citizens, the public, private sectors, and academic sectors' participation, collaboration, and partnership play a vital role in successful smart cities. ICT has the potential to facilitate citizen's participation and engagement in smart city decision-making, development, and maintenance. Citizen engagement is done in many ways, including through online platforms to submit feedback, social media to share information and propose plans for smart cities, mobile apps to complain and use smart services provided,

**Table 3  Elements of smart cities.**

| Infrastructure stability |
| --- |
| Policy, education, training, awareness, and technology |
| Educate engage residential and commercial sectors. |
| Technical and financial viability, affordability, localization and social impact. |
| Environmental initiatives. Effective and highly functional public transportation. Society culture |
| Population Security and privacy of the public |
| Direct effect on people and reducing expenditure |
| Data is the foundation, facilitating IoT connectivity, modular Infrastructure, security and distributed data processing. |
| Smart city technologies should be designed with sustainability in mind to ensure that they are environmentally friendly and resource-efficient. |
| How do we achieve the new technology projects if the city already exists and is occupied by people. |
| Cities need to make sure that the benefits of smart city technology outweigh the costs. |
| Open data and interconnectivity. |
| Consider citizen needs, data privacy and security, interoperability and scalability, collaboration and partnerships, and funding and financing |

in-person meetings with groups of citizens to get feedback, and citizen advisory boards for active engagement with diverse groups of citizens (*Lee et al., 2023*). There are many participation tools used to engage citizens in smart cities: ad hoc e-platforms, mobile apps, living labs, social media, datasets, gamification, sensors (IoT), exhibitions, workshops, and open data (*Horrigan, 2019*).

Question 7: How do you think smart cities can best address issues related to affordability and equity?

When designing and implementing smart cities, affordability and equity are two important considerations. The ability of residents to access and benefit from smart city services is referred to as affordability. Equity refers to the equitable distribution of smart city development benefits among all segments of the population. Focusing on low-cost solutions with high impact is one way to make smart cities more affordable. Using smart technologies, cities can improve public transportation, making it more affordable and efficient for residents to travel. In addition, cities can use smart technologies to reduce energy consumption and waste, saving residents money. Subsidizing access to smart city services like public Wi-Fi, transportation, and energy affordability can be created for citizens (*Akande, Cabral & Casteleyn, 2020*). In this study, experts cite various initiatives to achieve affordability and equity. According to expert opinion, affordable housing, transportation, and education are the most important issues. Portland, Oregon's smart city PDX prioritizes projects that reduce inequalities for people who have been left behind in city. The City of Philadelphia's SmartCityPHL Roadmap aims to involve the general public in the planning process (*Gracias et al., 2023*). Achieving affordability is a long term process by initiating careful urban planning with long term vision. Leveraging technological advancements in data-driven decision making, modular construction more importantly fostering innovation and collaboration. Experts suggest multiple aspects to focus on to

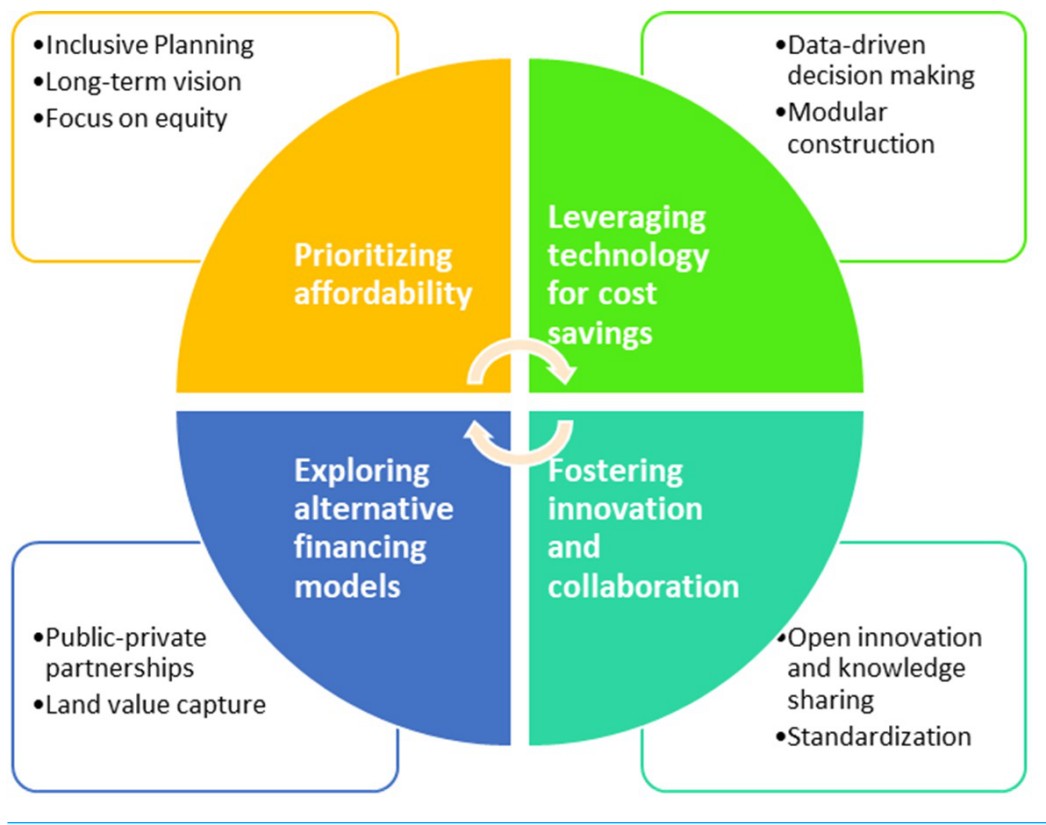

**Figure 6** Smart cities affordability aspects.

realize the vision of affordability in Fig. 6 these aspects are illustrated. Jordon urban policy initiated a roadmap to provide affordability by focusing on above mentioned aspects (*UN-Habitat, 2024*).

Question 8: What are the most important areas of research and development that should be prioritized in the field of smart cities?

Determining the smart city's research and development priorities is essential for achieving a set of objectives regarding resource allocation and social benefit. Experts who participated in our survey have pointed-out important areas of research and development to give high priority. Their suggestions are categorized as given in the Table 4. Ismagilova et al. identifies key research themes within the information systems perspective of smart cities, including data management, cybersecurity, interoperability, user-centered design, evaluation, and governance. Further, they have acknowledged the limitations in current research such as citizen participation and digital divides (*Ismagilova et al., 2019*). *Sánchez-Corcuera et al. (2019)* reviewed the open research challenges of smart cities and highlighted the research opportunities which includes use of AI and encryption without compromising the performance, application of fog and edge computing for local computation, Integration of Blockchain technology to smart cities architecture to improve trust between the stakeholder of smart cities (*Sánchez-Corcuera et al., 2019*). *Javed et al. (2022)* identified open issues, challenges, and future research potential in application of

| Table 4 | Research and Development priorities. |
| --- | --- |
| **Information security and privacy** | |
| Data governance and management | |
| Climate and Sustainability | |
| Multidisciplinary applied research | |
| Infrastructure | |
| Transportation | |
| Citizen engagement and participation | |
| Resilience and disaster management. | |

technologies like 6G networks, big data 2.0, Wi-Fi 7, Industry 5.0, advanced robotics and cyber security in smart cities. Application of deep learning techniques to handle the huge amount of data generated by IoT in smart cities is a significant research direction, which is studied by *Bhattacharya et al. (2022)* in their article on deep learning for future smart cities (*Bhattacharya et al., 2018*). The practice of safeguarding systems and data against unauthorized use, disclosure, disruption, alteration, or destruction is known as information security. Collecting and analyzing data is critical for smart cities, which operate on a complex network of interconnected devices and systems.

## DISCUSSION AND IMPLICATION

To understand the potential of smart technologies in developing countries, investigating the factors that may adopt and accept smart city development in Jordan is essential. The definition of a smart city plays a crucial role in reviewing the argument about smart cities' ethical implications (*Ziosi et al., 2022*). One of the best investments for the future in Jordan is adopting smart cities. One of the crucial practical implications of this adoption in Jordan is that all required information is available in an automated mode. This information can be utilized for the decision-making process through safe and manual programming in all sectors (health, education, economic, and scientific) (*Patel & Doshi, 2019*). However, technology and information are essential to successfully implementing smart cities. Several implications may exist as consequences related to smart cities. A main ethical implication of smart cities concerns the surveillance of their citizens (*Ziosi et al., 2022*). On the other hand, smart cities have several social implications like connected citizens and end-user privacy. Smart cities cover three pillars: environmental, social, and technical. From the environmental perspective, smart cities may reduce the undesired environmental effects that are intended to be environmentally friendly. On the other hand, from the technical perspective, smart cities use a wide range of technologies. From a social view, smart cities use the creativity of populations to improve digital communications. Social implication significantly impacts smart cities, where the health level and education can be improved (*Gracias et al., 2023*). Further, the community makes decisions that may affect the lives and the social environment. However, people are a key element in smart cities.

Jordan seeks to achieve sustainable development in the long run thus, major transformations to take benefit of smart technologies are observed. However, different

factors may significantly moderate the relationship between the ease of use, social influence and behavioral intention.

Overall, the implementation of smart cities in Jordan presents a transformative approach to urban development, offering both significant advantages and notable challenges. This study strengthens the idea that the integration of technology and data can achieve several objectives in civilized cities in order to become smart cities. The professional urban and technology designers are trying to create competences, improve economic development. Moreover, valid efforts were made in order to grow sustainability, and improve the overall quality of life for people living and working within the modern cities environment. One of the more significant findings to emerge from this study understands the potential merits and limitations of the smart cities and their future. Furthermore, this study investigates the future of smart cities in respect of developing countries. This investigation derived from the field experts' views and opinions. The responses were collected from several domains of smart cities such as smart governance, Education, Healthcare, communication, transportation, security, energy, and sustainability.

The development of smart cities in Jordan heralds a new dawn in urban innovation, bringing with it both immense opportunities and distinct challenges. On the one hand, the embrace of smart governance has ushered in an era of enhanced transparency and citizen engagement. Jordan's adoption of smart governance has paved the way for a more transparent, efficient, and responsive administration. Through digital platforms and e-services, citizens have gained easier access to essential services and can engage more actively with their local governments. This fosters a sense of involvement and accountability. The development and implementation of smart cities in Jordan have shown noteworthy significance in various sectors. In terms of smart governance, there has been a noticeable enhancement in transparency, efficiency, and public participation, leading to better decision-making and accountability.

Digital advancements in education promise a more inclusive learning environment, transcending geographical boundaries. Moreover, the integration of advanced technologies in the educational sector has reshaped the learning environment in Jordan. With the advent of e-learning platforms and digital classrooms, students across the country can access quality education, irrespective of their geographical location. This democratization of education is pivotal for national development. The education sector has benefited from smart solutions, introducing a more personalized learning experience and bridging the gap between urban and rural educational facilities. Healthcare has seen improved patient care through telemedicine, real-time monitoring, and data-driven approaches. Energy management has become more efficient with the incorporation of smart grids and renewable sources, reducing wastage and promoting sustainability. Advanced communication networks have fostered better connectivity, ensuring seamless interaction and information flow. The transportation sector has witnessed reduced traffic congestion and improved public transport systems through real-time tracking and smart traffic management. Security has been strengthened with advanced surveillance, predictive policing, and rapid response mechanisms.

The focus on smart energy solutions, such as smart grids and renewable energy sources, has enhanced the sustainability and efficiency of Jordan's energy sector. This not only addresses the country's energy needs but also positions Jordan as an advocate for environmental sustainability in the region. Sustainability initiatives, central to smart cities, have guided Jordan towards eco-friendly practices.

Smart healthcare solutions in Jordanian cities have optimized patient care and management. Telemedicine, electronic health records, and digital diagnostic tools have not only improved the quality of healthcare but also its accessibility, especially for those in remote areas. The healthcare sector, with its smart solutions, is poised to offer improved patient care, while the focus on sustainable energy solutions signifies a commitment to environmental responsibility. However, these advancements come with their set of challenges. The hefty investments required for technological infrastructure, the potential widening of the digital divide, concerns over data privacy, and the cultural shift towards a digital-first approach are hurdles that Jordan must address. Balancing these opportunities and challenges is crucial as Jordan strides forward in its smart city vision.

## CONCLUSION

In conclusion, while the move towards smart cities in Jordan heralds a new era of modernization and efficiency, it is essential to address the associated challenges head-on. With a balanced approach, Jordan can harness the full potential of smart cities for the betterment of its citizens and the broader region. The evidence from this study suggests that one of the best investments for the future in Jordan is adopting smart cities. However, with these advancements come certain limitations. While smart cities in Jordan offer a promising future with numerous benefits across various sectors, it is essential to address the associated challenges to ensure a holistic and inclusive development. The initial financial investment required for smart city infrastructure can be substantial. A digital divide needs to be addressed to ensure that all citizens have equal access to the benefits of smart technologies. Concerns regarding data privacy and potential misuse of personal information remain a challenge. The rapid pace of technological advancement means that infrastructure and systems need constant updating, which can be resource-intensive. Moreover, there is a need for continuous training and education for both officials and the public to adapt to and efficiently utilize these smart solutions. The following challenged the adoption of smart cities in Jordan:

Infrastructure and investment: The transformation to a smart city requires significant investment in infrastructure and technology. For a developing country like Jordan, mobilizing the necessary resources can be challenging.

Digital divide: While smart city initiatives aim to bridge gaps, they can inadvertently exacerbate the digital divide. Those without access to technology or the internet might feel further marginalized.

Data privacy and security: With the increased reliance on digital platforms, concerns about data privacy and cybersecurity arise. Ensuring the protection of citizens' data is paramount, and the country needs robust mechanisms to address potential breaches.

Cultural and behavioral adaptation: the shift to a digital-first approach might encounter resistance from certain sections of the population, especially those accustomed to traditional methods. Ensuring a smooth transition requires significant efforts in awareness and training. The adoption of smart cities in Jordan presents good opportunities for economic growth, sustainability, and improved quality of life. However, to realize the potential of smart cities in Jordan, several challenges need to be addressed. These consist of developing a comprehensive policy and regulatory framework, supporting citizen engagement, supporting digital inclusion, ensuring data privacy, improving skills, raising collaboration, prioritizing sustainability and resilience, and assessing economic and social impacts. By implementing the recommended strategies and approving the future directions, Jordan can overcome the optional challenges.

In collaboration with different stakeholders, the government should continue to prioritize the development of smart cities by adopting innovation, evolving technologies, and supporting the environment for sustainable development.

The successful implementation of smart cities in Jordan will take the kingdom to another level and be an international leader in sustainability, modern technology, and innovation. The successful adoption will also lead to improved quality of life for citizens, increase economic expansion, and develop environmental sustainability. However, this success needs a comprehensive approach integrating sustainable and technological development. Jordan can build technological, resilient, and sustainable cities. Furthermore, this success needs a full commitment for the long term. As mentioned before, the best investment for the future in Jordan is to adopt smart cities. Although smart cities in Jordan offer a favorable future, they face several challenges. The required financial investment for smart city infrastructure can be significant accordingly, the transformation to a smart city requires significant investment in technology and infrastructure. The fast technology development requires constant updating of the system and infrastructure. Data privacy is still a crucial challenge. Confirming the protection of citizens' data is vital. Jordan needs robust tools to address potential gaps. On the other hand, continuous valuable training for the public and private is needed. Finally, culture and behavioral adaptation is a significant challenge, where the shift to a digital approach may face resistance thus, significant efforts are needed as awareness and training sessions.

### Funding
This study is supported via funding from Prince Sattam bin Abdulaziz University project number (PSAU/2024/R/1445). The funders had no role in study design, data collection and analysis, decision to publish, or preparation of the manuscript.

### Grant Disclosures
The following grant information was disclosed by the authors:
Prince Sattam bin Abdulaziz University project: PSAU/2024/R/1445.

## Competing Interests

The authors declare there are no competing interests.

## Author Contributions

- Muneer Nusir conceived and designed the experiments, performed the experiments, analyzed the data, performed the computation work, prepared figures and/or tables, authored or reviewed drafts of the article, and approved the final draft.
- Mohammad Alshirah conceived and designed the experiments, performed the experiments, analyzed the data, authored or reviewed drafts of the article, and approved the final draft.
- Sahar A.L. Mashaqbeh conceived and designed the experiments, performed the experiments, analyzed the data, authored or reviewed drafts of the article, and approved the final draft.
- Mohammed Yousuf uddin conceived and designed the experiments, performed the experiments, analyzed the data, prepared figures and/or tables, authored or reviewed drafts of the article, and approved the final draft.
- Sultan Ahmad conceived and designed the experiments, authored or reviewed drafts of the article, and approved the final draft.
- Sana Fakhfakh performed the computation work, prepared figures and/or tables, authored or reviewed drafts of the article, proofreading, and approved the final draft.

## Data Availability

The raw data is available in the Supplemental File.

## Supplemental Information

Supplemental information for this article can be found online at http://dx.doi.org/10.7717/peerj-cs.2061#supplemental-information.

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
