# Peer review of "The Delphi method to analyze the expert views on possible futures of the smart city adoption and development in developing countries: the case of Jordan"

_PeerJ Computer Science, doi:10.7717/peerj-cs.2061_

## Round 0.1 · original submission · Major Revisions

Please adapt the manuscript to the reviewers' comments.

**Language Note:** PeerJ staff have identified that the English language needs to be improved. When you prepare your next revision, please either (i) have a colleague who is proficient in English and familiar with the subject matter review your manuscript, or (ii) contact a professional editing service to review your manuscript. PeerJ can provide language editing services - you can contact us at [email protected] for pricing (be sure to provide your manuscript number and title). – PeerJ Staff

Reviewer 1 ·

Basic reporting

The article is well written in English. The references included are good. However is necessary the addition of more references around the questions ( and answers) established in the paper. For example in the comments about the answers to the 8 questions in my opinion authors need more comments about papers in literature. In case of answers to question 8 there are no citations of previous work developed.

Experimental design

questions (8) are well written en very clear. In my opinion there is no problem with the clariry of the questionarie

Validity of the findings

As mentioned the results are congruent with studies developed. The paper analyzed the case of Jaornad. However it is necessary in the literature works about smart cities in at least one different country

Additional comments

Please:
1) add more studies in the literature review related with the discussion of each of the 8 questions. I mean, when you comment each answer for each question add more discussion with more papers. For example in ypur comment to question 8 there is no references of the literature
2) You should comment in the literature review an discussion cases of development of smart cities from other countries ( at least one)

Reviewer 2 ·

Basic reporting

The language is coherent, and the structure seems to follow the conventions of a formal paper. The text discusses smart healthcare solutions in Jordanian cities, addressing both the positive aspects and challenges associated with the adoption of smart city technologies.

The paper do not explicitly mention hypotheses or present clear results. A scientific or research paper typically includes a section on hypotheses, methodologies, and results. Ensuring that these elements are explicitly stated and connected is crucial for the clarity of the research.

While the paper mentions challenges and benefits of smart healthcare solutions, there seems to be a need for a more structured organization. Scientific papers usually follow a specific format with sections like Introduction, Literature Review, Methodology, Results, Discussion, and Conclusion. Clarifying the structure can enhance the paper's coherence.

Follow a standard scientific paper structure with sections such as Introduction, Literature Review, Methodology, Results, Discussion, and Conclusion. This will enhance the paper's coherence and make it more reader-friendly.

Experimental design

Provide more technical details, especially in the Methodology and Results sections. Clearly explain the methodologies used and reference relevant prior literature to contextualize the research.

Include relevant figures, tables, or raw data to support the points discussed. Visual aids can enhance the understanding of the research findings and provide a basis for the conclusions.

Validity of the findings

The conclusion section could be more explicit in summarizing the key findings and addressing how the challenges mentioned can be overcome. A well-crafted conclusion ties together the entire paper and provides insights for future research or practical applications.

External reviews were received for this submission. These reviews were used by the Editor when they made their decision, and can be downloaded below.

---

## Round 0.2 · accepted · Accept

Dear Authors,

Congratulations on your acceptance.

Reviewer 1 ·

Basic reporting

All points are included in the last version of authors. The paper should be accepted. The literature review is much better

Experimental design

All points are included in the last version of authors. The paper should be accepted. Rigous reserach is developed

Validity of the findings

All point are included in the last version of authors. The paper should be accepted

Additional comments

The paper should be accepted

External reviews were received for this submission. These reviews were used by the Editor when they made their decision, and can be downloaded below.